# Medical history predicts phenome-wide disease onset and enables the rapid response to emerging health threats

Jakob Steinfeldt[1,2,3,4,5,18], Benjamin Wild [5,6,18], Thore Buergel[5,6,18], Maik Pietzner [3,7,8], Julius Upmeier zu Belzen [6], Andre Vauvelle[9], Stefan Hegselmann[10,11], Spiros Denaxas[9,12,13,14], Harry Hemingway [9,13,14], Claudia Langenberg [3,7,8], Ulf Landmesser[1,2,4,15,16,19], John Deanfield[5,19] & Roland Eils [6,17,19] ✉

The COVID-19 pandemic exposed a global deficiency of systematic, data-driven guidance to identify high-risk individuals. Here, we illustrate the utility of routinely recorded medical history to predict the risk for 1741 diseases across clinical specialties and support the rapid response to emerging health threats such as COVID-19. We developed a neural network to learn from health records of 502,489 UK Biobank participants. Importantly, we observed discriminative improvements over basic demographic predictors for 1546 (88.8%) endpoints. After transferring the unmodified risk models to the All of US cohort, we replicated these improvements for 1115 (78.9%) of 1414 investigated endpoints, demonstrating generalizability across healthcare systems and historically underrepresented groups. Ultimately, we showed how this approach could have been used to identify individuals vulnerable to severe COVID-19. Our study demonstrates the potential of medical history to support guidance for emerging pandemics by systematically estimating risk for thousands of diseases at once at minimal cost.

The early phase of the COVID-19 pandemic exposed a global deficiency in delivering systematic, data-driven guidance for individual patients and healthcare providers with critical implications for pandemic preparedness. The assessment of an individual's risk for future disease is central to guiding preventive interventions, early detection of disease, and the initiation of treatments. However, bespoke risk scores are only available for a subset of common diseases[1–4], leaving healthcare providers and individuals with little to no guidance on most relevant

[1]Department of Cardiology, Angiology and Intensive Care Medicine, Deutsches Herzzentrum der Charité (DHZC), Berlin, Germany. [2]Charité – Universitätsmedizin Berlin, corporate member of Freie Universität Berlin and Humboldt Universität zu Berlin, Klinik/Centrum, Berlin, Germany. [3]Computational Medicine, Berlin Institute of Health (BIH), Charite - University Medicine Berlin, Berlin, Germany. [4]Friede Springer Cardiovascular Prevention Center@Charite, Charite - University Medicine Berlin, Berlin, Germany. [5]Institute of Cardiovascular Sciences, University College London, London, UK. [6]Center for Digital Health, Berlin Institute of Health (BIH), Charite - University Medicine Berlin, Berlin, Germany. [7]MRC Epidemiology Unit, Institute of Metabolic Science, University of Cambridge, Cambridge, UK. [8]Precision Health University Research Institute, Queen Mary University of London and Barts NHS Trust, London, UK. [9]Institute of Health Informatics, University College London, London, UK. [10]Institute for Medical Engineering and Science, Massachusetts Institute of Technology, Massachusetts, USA. [11]Pattern Recognition and Image Analysis Lab, University of Münster, Münster, Germany. [12]British Heart Foundation Data Science Centre, London, UK. [13]Health Data Research UK, London, UK. [14]National Institute for Health Research, Biomedical Research Centre at University College London Hospitals, London, UK. [15]Berlin Institute of Health (BIH), Charite - University Medicine Berlin, Berlin, Germany. [16]DZHK (German Centre for Cardiovascular Research), Partner Site Berlin, Berlin, Berlin, Germany. [17]Health Data Science Unit, Heidelberg University Hospital and BioQuant, Heidelberg, Germany. [18]These authors contributed equally: Jakob Steinfeldt, Benjamin Wild, Thore Buergel. [19]These authors jointly supervised this work: Ulf Landmesser, John Deanfield, Roland Eils. ✉e-mail: roland.eils@bih-charite.de

diseases. Even for diseases with established risk scores, little consensus exists on which score to use and associated physical or laboratory measurements to obtain, leading to highly fragmented practice in routine care[5]. Importantly, in the early phases of emerging pandemics such as COVID-19, it is necessary to allocate sparse resources, but risk scores to identify vulnerable subpopulations are not available due to the lack of available data.

At the same time, most medical decisions on diagnosis, treatment, and prevention of diseases are fundamentally based on an individual's medical history[6]. With the widespread digitalization, this information is routinely collected by healthcare providers, insurance, and governmental organizations at a population scale in the form of electronic health records[7–12]. These readily accessible records, which include diseases, medications, and procedures, are potentially informative about future risk trajectories, but their potential to improve medical decision-making is limited by the human ability to process and understand vast amounts of data[13].

To date, routine health records have been used to guide clinical decision-making with etiological[14–17], diagnostic[18,19], and prognostic research[15,16,20–22]. Existing efforts often extract and leverage known clinical predictors with new methodologies[19], augment them with additionally extracted data modalities such as clinical notes[23], or aim to identify novel predictors among the recorded concepts[14–17]. Prior work on the prediction of disease onset has mainly focused on single diseases, including dementia[15,24], cardiovascular conditions[23,25] such as heart failure[26] and atrial fibrillation[27,28]. In contrast, phenome-wide association studies (PheWAS) quantifying the associations of genetic variants with comprehensive phenotypic traits are emerging in genetic epidemiology[29,30]. While approaches have been developed for high-throughput phenotyping[31,32] and to extract information from longitudinal health records[33,34], no studies have investigated the predictive potential and potential utility over the entire human phenome. Consequently, the predictive information in routinely collected health records and its potential to systematically guide medical decision-making is largely unexplored.

Here, we examined the predictive potential of an individual's entire medical history and proposed a systematic approach for phenome-wide risk stratification. We developed, trained, and validated a neural network in the UK Biobank cohort[35] to estimate disease risk from routinely collected health records. Unlike alternative methods such as linear models or survival trees, which require separate models for each disease, our approach employs a multi-layer perceptron that predicts multiple endpoints concurrently, resulting in a significantly simplified model architecture. These endpoints include preventable diseases (e.g., coronary heart disease), diseases that are not currently preventable, but the early diagnosis has been shown to substantially slow down the progression and development of complications (e.g., heart failure), and outcomes that are currently neither entirely preventable nor treatable (e.g., death). They also include both diseases with risk prediction models recommended in guidelines and used in practice (e.g., cardiovascular diseases or breast cancer) as well as diseases without current risk prediction models (e.g., psoriasis and rheumatoid arthritis).

We evaluated our approach by integrating the endpoint-specific risk states estimated by the neural network in Cox Proportional Hazard models[36], investigating the phenome-wide predictive potential over basic demographic predictors, selected comorbidities, and established modifiable risk factors, and illustrating how phenome-wide risk stratification could benefit individuals by providing risk estimates, facilitating early disease diagnosis, and guiding preventive interventions. Furthermore, by externally validating in the All Of Us cohort[37], we show that our models can generalize across healthcare systems and populations, including communities historically underrepresented in biomedical research.

Finally, we assessed the potential of our approach to aid risk stratification for the primary prevention of cardiovascular disease and to respond to emerging health threats in the example of COVID-19. We then show that the risk states of pneumonia, sepsis & all-cause death can be used to calculate a combined severity risk score using primary and secondary care records available before the global spread of the COVID-19 pandemic. Our results demonstrate the currently unused potential of routine health records to guide medical practice by providing comprehensive phenome-wide risk estimates.

## Results

### Characteristics of the study population and integration of routine health records

This study is based on the UK Biobank cohort[35,38], a longitudinal population cohort of 502,489 relatively healthy individuals of primarily British descent, with a median age of 58 (IQR 50, 63) years, 54.4% biological females, 11% current smokers, and a median BMI of 26.7 (IQR 24.1, 29.9) at recruitment (Table 1 for detailed information). Individuals recruited between 2006 and 2010 were followed for a median of 13.8 years, resulting in ~ 6.8 M overall person-years on 1741 phenome-wide endpoints[39] with ≥100 incident events (>0.02% of individuals having the event in the observation time). We externally validated our findings in individuals from the All of Us cohort, a longitudinal cohort of 259,234 individuals with linked health records recruited from all over the United States. Individuals in the All of Us cohort are of diverse descent, with 45% of reportedly non-white ethnicity and 76% of groups historically underrepresented in biomedical research[37,40] and have a median age of 54 (IQR 38, 66) years with 61.0% biological females (see Table 1 for detailed information). Individuals were recruited from 2019 on and followed for a median of 3.1 years, resulting in ~ 840,574 person-years on 1519 endpoints.

Central to this study is the prior medical history, defined as the entirety of routine health records before recruitment. Before further analysis, we mapped all health records to the OMOP vocabulary. While most records originate from primary care and, to a lesser extent, secondary care (Supplementary Fig. 1a), the predominant record domains are drugs and observations, followed by conditions, procedures, and devices (Supplementary Fig. 1b). Interestingly, while rare medical concepts (with a record in <1% of individuals in the study population) are not commonly included in prediction models[21], they are often associated with high incident event rates (exemplified by the mortality rate in Supplementary Fig. 1c) compared to common concepts (a record present in ≥1% of the study population). For example, the concept code for "portal hypertension" (OMOP 34742003) is only recorded in 0.04% (193) of individuals at recruitment, but 50.8% (98 individuals) will die over the course of the observation period. Importantly, there are many distinct rare concepts, and thus 74.4% of individuals have at least one rare record before recruitment, compared with 78.8% for common records. In addition, 40.1% of individuals have ≥10 rare records compared with 45.1% for common records, and individuals have only slightly fewer rare than common records (Supplementary Fig. 1d).

After excluding very rare concepts (<0.01%, less than 50 individuals with the record in this study), we integrated the remaining 14,444 unique concepts (Supplementary Data 2) with a multi-task multi-layer perceptron (with 82.3 M parameters) to predict the phenome-wide onset of 1741 endpoints (Supplementary Data 1) simultaneously (Fig. 1a). For comparison, we also include additional comparisons with a linear baseline (with 25 M parameters, Supplementary Fig. 2), demonstrating superior performance at a minimal increase of complexity.

To ensure that our findings are generalizable and transferable, we spatially validate our models in 22 recruitment centers (Fig. 1b) across England, Wales, and Scotland. We developed 22 models, each trained

**Table 1 | The study population**

| | | UK Biobank | | | All Of Us | | | |
|---|---|---|---|---|---|---|---|---|
| | | Male, $N = 229,114$ | Female, $N = 273,375$ | Overall, $N = 502,489$ | Male, $N = 95,950$ | Female, $N = 158,025$ | Diverse/ Unknown, $N = 5259$ | Overall, $N = 259,234$ |
| Age (years) | | 58 (50, 64) | 57 (50, 63) | 58 (50, 63) | 57 (42, 68) | 52 (36, 64) | 58 (44, 67) | 54 (38, 66) |
| | Unknown | – | – | – | 2166 (2,3%) | 3717 (2,4%) | 120 (2,3%) | 6003 (2,3%) |
| Ethnicity | Asian | 5294 (2.3%) | 4588 (1.7%) | 9882 (2.0%) | 2584 (2.7%) | 4302 (2.7%) | 44 (0.8%) | 6930 (2.7%) |
| | Black | 3407 (1.5%) | 4654 (1.7%) | 8061 (1.6%) | 19,674 (21%) | 30,148 (19%) | 829 (16%) | 50,651 (20%) |
| | Mixed | 1105 (0.5%) | 1853 (0.7%) | 2958 (0.6%) | 16,500 (17%) | 34,919 (22%) | 478 (9.1%) | 51,897 (20%) |
| | White | 215,251 (95%) | 257,429 (96%) | 472,680 (95%) | 55,277 (58%) | 86,104 (54%) | 1182 (22%) | 142,563 (55%) |
| | Unknown | 3473 (1.5%) | 3861 (1.4%) | 7334 (1.5%) | 1915 (2.0%) | 2552 (1.6%) | 2726 (52%) | 7193 (2.8%) |
| Smoking status | Current | 28,610 (13%) | 24,367 (9.0%) | 52,977 (11%) | 13,973 (15%) | 13,660 (8.6%) | 649 (12%) | 28,282 (11%) |
| | Previous | 87,605 (38%) | 85,445 (31%) | 173,050 (35%) | 25,575 (27%) | 33,652 (21%) | 1232 (23%) | 60,459 (23%) |
| | Never | 111,463 (49%) | 162,051 (60%) | 273,514 (55%) | 46,193 (48%) | 99,502 (63%) | 2654 (50%) | 148,349 (57%) |
| | Unknown | 1436 (0.6%) | 1512 (0.7%) | 2948 (0.5%) | 10,209 (11%) | 11,211 (7.1%) | 724 (14%) | 22,144 (8.5%) |
| Body Mass Index | – | 27.3 (25.0, 30.1) | 26.1 (23.5, 29.7) | 26.7 (24.1, 29.9) | 28 (25, 32) | 29 (25, 35) | 29 (25, 34) | 29 (25, 34) |
| | Unknown | 1646 (0.7%) | 1458 (0.6%) | 3104 (0.6%) | 3772 (3.9%) | 7576 (4.8%) | 196 (3.7%) | 11,544 (4.5%) |
| Systolic Blood Pressure (mmHg) | – | 139 (129, 152) | 133 (121, 147) | 136 (125, 150) | 129 (119, 141) | 123 (112, 136) | 127 (115, 139) | 125 (114, 138) |
| | Unknown | 13,580 (5.9%) | 16,538 (6.0%) | 30,118 (6.0%) | 2166 (2.3%) | 3717 (2.4%) | 120 (2,3%) | 6003 (2,3%) |

Median (IQR); n (%).

on individuals from 21 recruitment centers at recruitment, randomly split into training and validation sets (Fig. 1c). We subsequently tested the models on individuals from the additional recruitment center unseen for model development for internal spatial validation. After checkpoint selection on the validation data sets and obtaining the selected models' final predictions on the individual test sets, the test set predictions were aggregated for downstream analysis (Fig. 1d). Subsequently, disease-specific exclusions of prior events and sex-specificity were respected in all downstream analyses. After development, the models were externally validated in the All of Us cohort[37].

**Routine health records stratify phenome-wide disease onset**
Central to the utility of any predictor is its potential to stratify risk. The better the stratification of low and high-risk individuals, the more effective targeted interventions and disease diagnoses are.

To investigate whether health records can be used to identify high-risk individuals, we assessed the relationship between the risk states estimated by the neural network for each endpoint and the risk of future disease (Fig. 2). For illustration, we first aggregated the incident events over the percentiles of the risk states for each endpoint and subsequently calculated ratios between the top and bottom 10% of risk states over the entire phenome (Fig. 2a). We found that fewer than 10% of the individuals had an incident hypertension diagnosis in the observation window if they were estimated to be in the bottom risk percentile of the medical history, compared to more than 60% if they were estimated to be in the top risk percentile. Subsequently, the incident event ratio between the top and bottom deciles was ~ 8.23. Importantly, we found differences in the event rates, reflecting a stratification of high and low-risk individuals for almost all endpoints covering a broad range of disease categories and etiologies: For 1118 of 1171 endpoints (64.2%), we observed >10-times as many events for individuals in the top 10% of the predicted risk states compared to the bottom 10%. For instance, these endpoints included rheumatoid arthritis (Ratio ~ 15.7), ischemic heart disease (Ratio ~ 18.4), or chronic obstructive pulmonary disease (Ratio ~ 68.6). For 207 (11.9%) of the 1741 conditions, including abdominal aortic aneurysm (Ratio ~ 84.8), more than 80 times the number of individuals in the top 10% of predicted risk states had incident events compared to the bottom 10%. For 623 (35.8%)

endpoints, the separation between high and low-risk individuals was smaller (Ratio <10), which included hypertension (Ratio ~ 8.2) and anemia (Ratio ~ 7.4), often diagnosed earlier in life or precursors for future comorbidities. Notably, the ratios were >1 for all but two of the 1741 investigated endpoints, even though all models were developed in spatially segregated assessment centers. To illustrate how high-risk individuals differ from the moderate cases, we also provide additional ratios comparing the top 10% to individuals in the median 20% of the population. The complete list of all endpoints and corresponding statistics can be found in Supplementary Data 4.

In addition to the phenome-wide analysis of 1741 endpoints, we also provide detailed associations between the risk percentiles and incident event ratios (Fig. 2b) as well as cumulative event rates for up to 15 years (Fig. 2c) of follow-up for the top, median, and bottom percentiles for a subset of 24 selected endpoints. This set was selected to comprise actionable endpoints and common diseases with significant societal burdens, specific cardiovascular conditions with pharmacological and surgical interventions, as well as endpoints without established tools to stratify risk to date. To exemplify the potential of our approach, among individuals in the top risk decile for heart failure, 6738 (13.48%) experienced an event, in contrast to 146 (0.29%) individuals in the bottom decile, resulting in a risk ratio of 46.15 (Fig. 2a, b and Supplementary Data 4). Consequently, those at high risk of heart failure could be prioritized for echocardiographic screening and, if necessary, prescribed effective guideline-directed medical therapy. Similarly, individuals with a high risk of developing COPD - where the top 10% face over 68 times the risk compared to the bottom 10% - may be considered for spirometry, an approach already established in the CAPTURE trial[41]. If confirmed, they could benefit from interventions such as long-acting bronchodilators. As a third example, a high-risk estimate for less common diseases, such as multiple sclerosis (risk ratio ~ 13.3), could further support referring individuals to a specialist and potentially shorten the often extensive patient journey before a final diagnosis is reached.

In summary, the disease-specific states stratify the risk of onset for all 1741 investigated endpoints across clinical specialties. This indicates that routine health records provide a large and widely unused potential for the systematic risk estimation of disease onset in the general population.

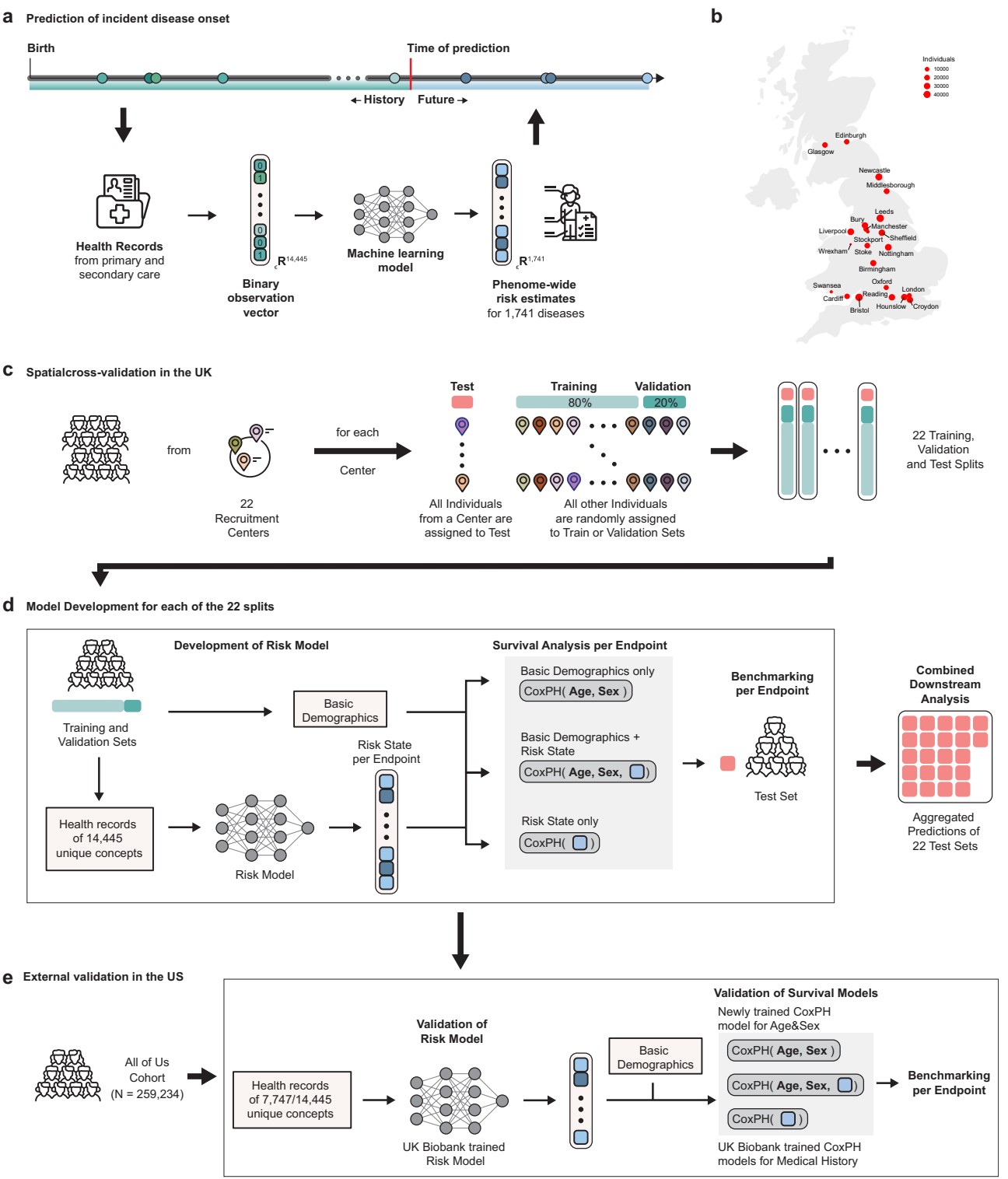

**Fig. 1 | Overview of the study. a** The medical history captures encounters with primary and secondary care, including diagnoses, medications, and procedures (ideally) from birth. Here we train a multi-layer perceptron on data before recruitment to predict phenome-wide incident disease onset for 1741 endpoints. **b** Location and size of the 22 assessment centers of the UK Biobank cohort across England, Wales, and Scotland. **c** To learn risk states from individual medical histories, the UK Biobank population was partitioned by their respective assessment center at recruitment. **d** For each of the 22 partitions, the Risk Model was trained to predict phenome-wide incident disease onset for 1741 endpoints. Subsequently, for each endpoint, Cox proportional hazard (CPH) models were developed on the risk states in combination with sets of commonly available predictors to model disease risk. Predictions of the CPH model on the test set were aggregated for downstream analysis. **e** External validation in the All of US cohort. After mapping to the OMOP vocabulary, we transferred the trained risk model to the All of Us cohort and calculated the risk state for all endpoints. To validate these risk states, we compared the unchanged CPH models developed in the UK Biobank with refitted CPH models for age and sex. Source data are provided. The Icons are made by Freepik from www.flaticon.com.

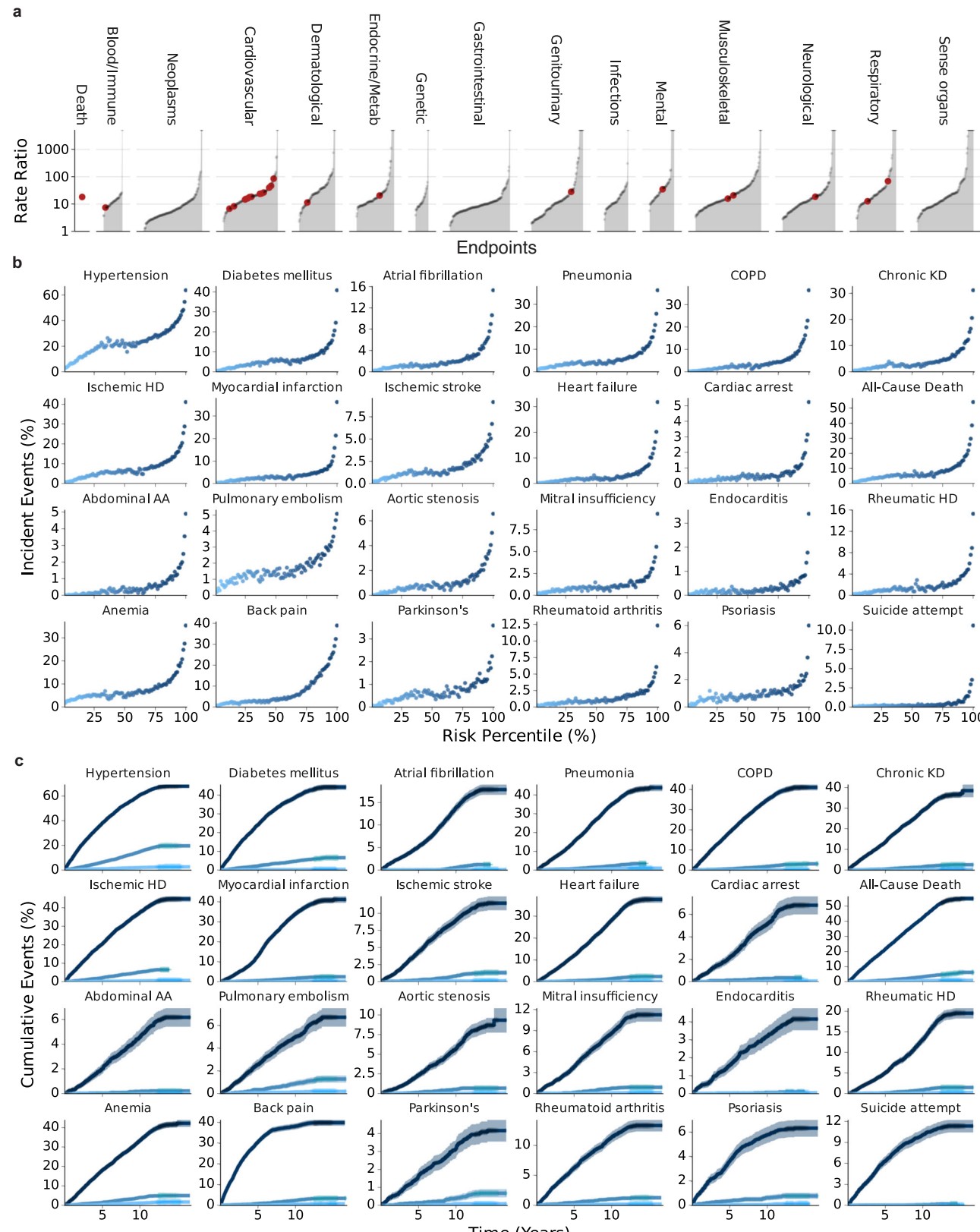

**Fig. 2 | Routine health records stratify phenome-wide disease onset. a** Ratio of incident events in the Top 10% compared with the Bottom 10% of the estimated risk states. Event rates in the Top 10% are higher than in the Bottom 10% for all but one of the 1741 investigated endpoints. Red dots indicate 24 selected endpoints detailed in Fig. 2b. To illustrate, 1053 (2.10%) individuals in the top risk decile for cardiac arrest experienced an event compared with only 59 (0.12%) in the bottom decile, with a risk ratio of 17.85. **b** Incident event rates for each medical history risk percentile (if medical history was available) for a selection of 24 endpoints. **c** Cumulative event rates with 95% confidence intervals for the Top 1%, median, and Bottom 1% of risk percentiles in (**b**) over 15ys. Statistical measures were derived from 502,489 individuals. Individuals with prevalent diseases were excluded from the endpoints-specific analysis. Source data are provided.

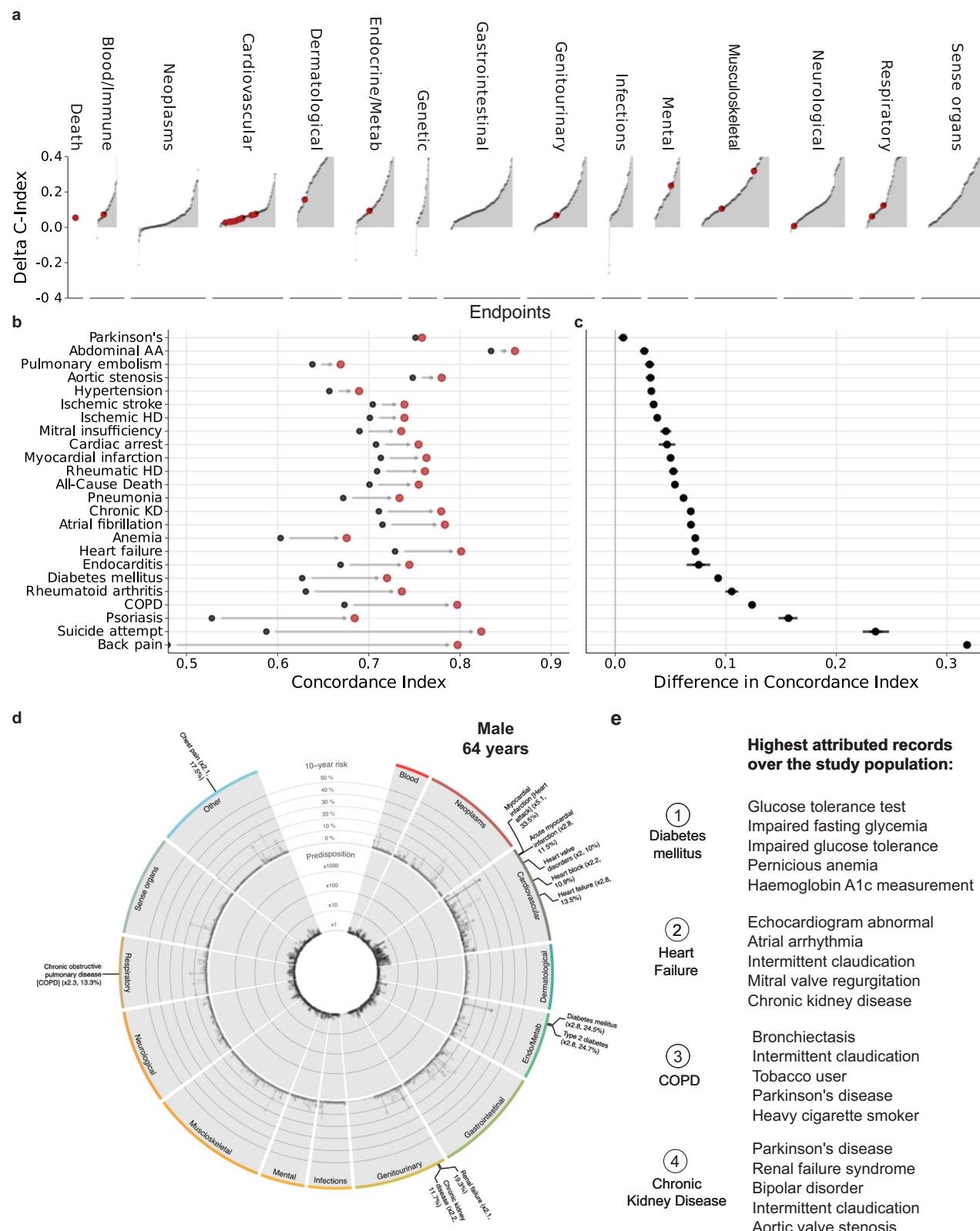

### Discriminative performance indicates potential utility

While routine health records can stratify incident event rates, this does not prove utility. To test whether the risk state derived from the routine health records could provide utility and information beyond ubiquitously available predictors, we investigated the predictive information over age and biological sex, selected comorbidities from the Charlson Comorbidity Index[42], and established modifiable risk factors from the AHA ASCVD pooled cohort equation[3]. We modeled the risk of disease onset using Cox Proportional-Hazards (CPH) models for all 1741 endpoints, which allowed us to estimate adjusted hazard ratios (denoted as HR in Supplementary Data 6) and 10-year discriminative improvements (indicated as Delta C-index in Fig. 3a).

**Fig. 3 | Discriminative performance indicates potential utility. a** Differences in discriminatory performance quantified by the C-Index between CPH models trained on Age + Sex and Age + Sex + MedicalHistory for all 1741 endpoints. We found significant improvements over the baseline model (Age + Sex, age, and biological sex only) for 1546 (88.8%) of the 1741 investigated endpoints. Red dots indicate selected endpoints in Fig. 3b. **b** Absolute discriminatory performance in terms of C-Index comparing the baseline (Age + Sex, black point) with the added routine health records risk state (Age + Sex + RiskState, red point) for a selection of 24 endpoints. **c** The direct C-index differences for the same models. Dots indicate medians and whiskers extend to the Bonferroni-corrected 95% confidence interval for a distribution bootstrapped over 100 iterations. **d** Example of individual predicted phenome-wide risk profile. Predisposition (10-year risk estimated by Age + Sex + RiskState compared to risk estimated by Age + Sex alone) is displayed in the inner circle, and absolute 10-year risk estimated by Age + Sex + RiskState can be found in the outer circle. Labels indicate endpoints with a high individual predisposition (>2 times higher than the Age + Sex-based reference estimate) and absolute 10-year risk >10%. **e** Top 5 highest attributed records for selected endpoints. Statistical measures were derived from 502,489 individuals. Source data are provided.

We found significant improvements over the baseline model (age and biological sex only) for 1546 (88.8%) of the 1741 investigated endpoints (Fig. 3 and Supplementary Data 5). For many of these endpoints, the discriminative improvements were considerable (Delta C-Index Q25%: 0.082, Q50: 0.101, Q75: 0.157). We found significant improvements for all of the 24 highlighted endpoints (indicated in Fig. 2a), with large increases for the prediction of back pain (Delta C-Index: + 0.318 (0.314, 0.324)), suicide attempts (Delta C-Index: + 0.236 (0.217, 0.253)), psoriasis (Delta C-Index: + 0.157 (0.144, 0.173)), all-cause mortality (Delta C-Index: + 0.054 (0.051, 0.058)) and chronic obstructive pulmonary disease (Delta C-Index: + 0.124 (0.118, 0.129)). In contrast, we did not find significant improvements in the prediction of 182 (10.5%) of the 1741 endpoints, including, e.g., multiple myeloma (Delta C-Index: − 0.006 (− 0.021, 0.009)). or even deteriorations in the prediction of 13 (0.7%) of the endpoints, including neoplasm like breast cancer (Delta C-Index: − 0.003 (− 0.005, − 0.001)) and gastrointestinal cancer (Delta C-Index: − 0.004 (− 0.007, − 0.001)).

We also present a comparison between our approach and the Charlson Comorbidity Index's[42] predictive performance, both of which can be automated. In addition, we compare our method to the well-established ASCVD predictors, which are widely accessible but require an additional blood draw. Notably, incorporating the comorbidities from the Charlson Comorbidity Index enhances the discriminative capacity beyond age and sex; however, adding medical history proves to be significantly more effective in improving performance (Supplementary Fig. 3 and Supplementary Data 5). Likewise, while supplementing ASCVD predictors to age and sex augments the performance for most endpoints, it remains inferior to the combination of age, sex, and medical history alone. Incorporating the medical history alongside the comorbidities or ASCVD predictors further improves the predictive performance for the vast majority of endpoints (AgeSex + Comorbidities augmented by the MedicalHistory: + 1415/1741 (81.2%), ASCVD + MedicalHistory: + 1431/1741 (82.2%)), demonstrating complementary nature of these information sources.

For illustration, we also present individual phenome-wide risk profiles (Fig. 3c and Supplementary Figs. 4a, b, 5a, b). The risk profiles varied substantially in the predispositions relative to the age and sex reference (the inner circle, see methods for details) and the absolute 10-year risk estimates (the outer circle). The first individual (Fig. 3c), a 64-year-old man, is predicted to be at a particularly high 10-year risk of metabolic, cardiovascular, respiratory, and genitourinary conditions, including diabetes mellitus (24.5%), heart failure (13.5%), COPD (13.3%), and chronic kidney disease (11.7%). Increased risk of neoplastic, dermatological, and musculoskeletal conditions was not predicted by the prior health records of this individual. In contrast, another individual, a 46-year-old woman (Supplementary Fig. 4b), is not estimated at increased cardiovascular risk but conversely to have almost 20x the risk for suicide ideation and attempt or self-harm compared to the reference group.

Importantly, the model performance is robust to the removal of recent information, indicating that the model effectively incorporates both the individuals' long-term medical history and recent interactions with the healthcare system in order to predict future disease onset (Supplementary Fig. 6). We provide Shapley attributions[43] for the most important records (Fig. 3d and Supplementary Figs. 4c, 5c) and all records for the 24 highlighted endpoints (Supplementary Data 9) in the study population, enhancing the interpretability of our findings.

These findings indicate that health records contain substantial predictive information over established predictors for the majority of disease endpoints from across clinical specialties.

## Predictive models can generalize across healthcare systems and populations

While our findings indicate potential utility in the UK Biobank, health records vary substantially across healthcare systems and over time due to differences in medical and coding practices ("distribution shift") and underlying differences in the populations. Thus, predictive models can fail to learn robust and generalizable information[44–46].

To better understand the generalizability across different healthcare systems, we predicted risk states and absolute risk estimates for all individuals in the All of Us cohort with linked medical records ($N = 259,234$; see Table 1). Importantly, we found significant improvements over the baseline model (age and biological sex only) for 1171 (77.1%) of the 1519 investigated endpoints with at least 100 incident events (Fig. 4a and Supplementary Data 8), replicating 1115/1414 (78.9%) of all significant improvements in the UK Biobank (Fig. 4b and Supplementary Data 8). Generally, larger improvements in the UK Biobank were replicated in the All of Us cohort. It is noteworthy that smaller improvements in the UK Biobank often corresponded to proportionately larger improvements in All of Us, while larger improvements in the UK Biobank were attenuated in All of Us (Fig. 4c).

As the risk states were largely derived from white, middle-aged, and generally affluent and healthy individuals from the UK, it was critical to validate the discriminative performance in diverse and historically underserved and underrepresented groups and ethnicities. Generally, we found comparable discriminative performances (Fig. 4d) and substantial benefits over basic demographic predictors (example of cardiac arrest in Fig. 4e) across all investigated groups.

To illustrate these improvements further, we replicated significant improvements for all of the 24 a priori selected endpoints, with improvements ranging from modest for hypertension (Delta C-Index: + 0.017 (0.013, 0.021)) and Parkinson's disease (Delta C-Index: + 0.032 (0.016, 0.046)) to substantial for, e.g., All-Cause Death (Delta C-Index: + 0.103 (0.092, 0.112), Pulmonary embolism (Delta C-Index: + 0.102 (0.084, 0.114)), and Cardiac arrest (Delta C-Index: + 0.17 (0.14, 0.194)) (Fig. 4f, g and Supplementary Data 8). Only for a subset of 78 (5.55%) significantly improved endpoints in the UK Biobank, the discriminative performance in All Of Us deteriorated significantly upon transferring the pre-trained medical history risk model and integrating the information beyond age and biological sex alone, including basal cell carcinoma (Delta C-Index: − 0.031 (− 0.051, − 0.012)), acne (Delta C-Index: − 0.147 (− 0.174, − 0.114)) and osteoporosis (Delta C-Index: − 0.019 (− 0.027, − 0.012)).

Taken together, our findings suggest that predictive models based on medical history can be generalized across health systems and are robust to diverse populations.

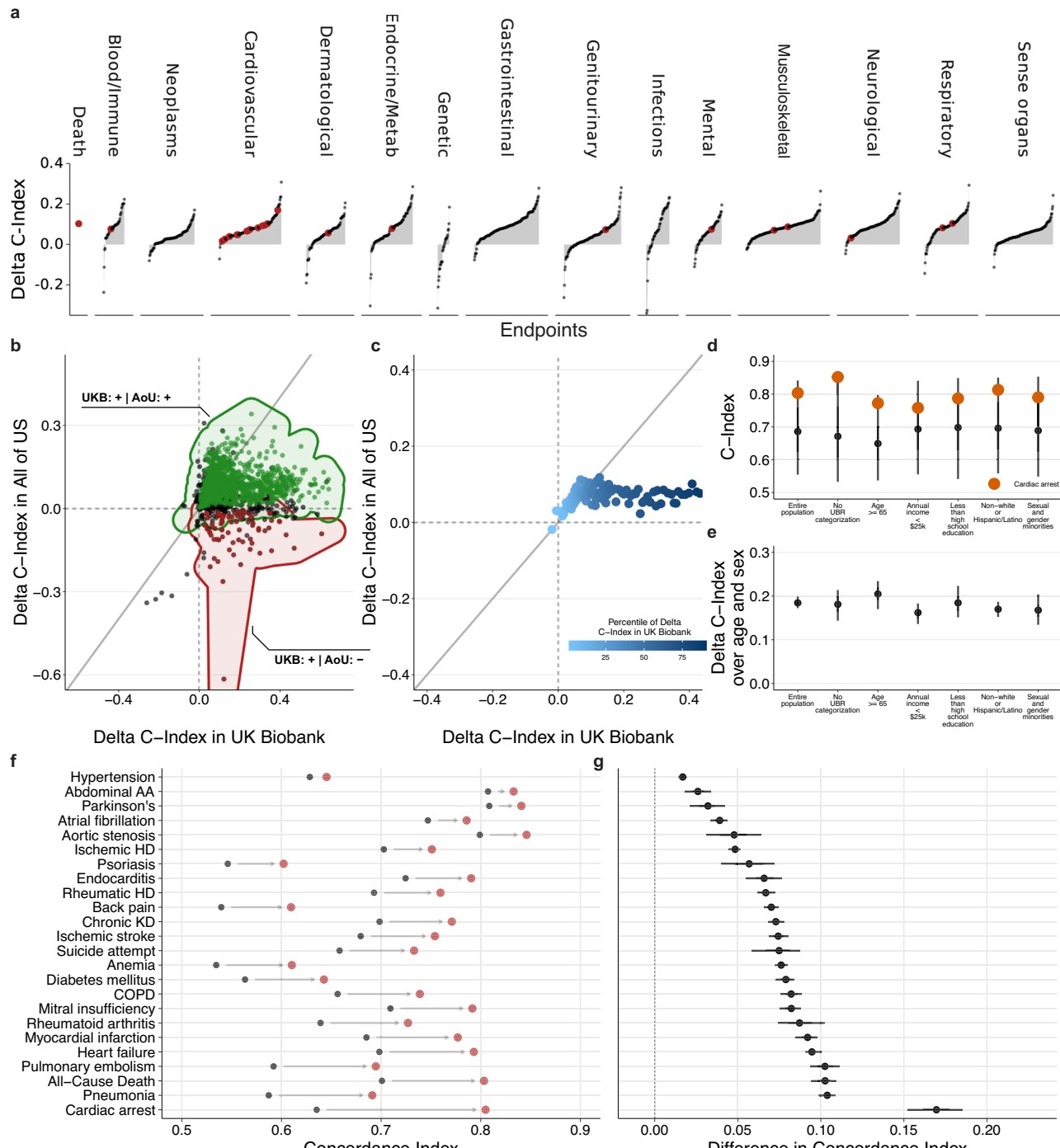

**Fig. 4 | Predictive models can generalize across healthcare systems and populations. a** External validation of the differences in discriminatory performance quantified by the C-Index between CPH models trained on age and biological sex and age, biological sex, and the risk state for 1519 endpoints in the All of Us cohort. We find significant improvements over the baseline model (age and biological sex only) for 1171 (77.1%) of the 1519 investigated endpoints. **b** Direct comparison of the absolute C-Index in the UK Biobank (*x*-axis) and the All Of Us cohort (*y*-axis). Significant improvements can be replicated for 1115 (78.9%, green points) of 1414 endpoints in the All Of Us cohort. **c** Comparison of mean delta C-Index per delta percentile (derived from the UK Biobank from the 1519 endpoints available in All Of Us). Improvements in the All Of Us cohort are consistent with the UK Biobank cohort: Small improvements in the UK Biobank tend to be larger in All Of Us, while large improvements in the UK Biobank tend to be attenuated in All Of Us. **d** Distribution of C-Indices for the 1519 investigated endpoints stratified by

communities historically underrepresented in biomedical research (UPD)[73]. Dots indicate medians and whiskers extend to the Bonferroni-corrected 95% confidence interval for a distribution bootstrapped over 100 iterations. **e** For the same groups, confidence intervals for the additive performance as measured by the C-Index compared to the baseline model. Dots indicate medians and whiskers extend to the Bonferroni-corrected 95% confidence interval for a distribution bootstrapped over 100 iterations. **f** Absolute discriminatory performance in terms of C-Index comparing the baseline (age and biological sex, black point) with the added routine health records risk state (red points) for a selection of 24 endpoints. **g** The differences in C-index for the same models. Statistical measures for UKB in (**b** and **c**) were derived from 502,489 individuals, and for AoU in (**a**–**g**) were derived from 259,234 individuals. Dots indicate medians and whiskers extend to the Bonferroni-corrected 95% confidence interval for a distribution bootstrapped over 100 iterations. Source data are provided.

### Predictions can support cardiovascular disease prevention and the response to emerging health threats

While comprehensive phenome-wide risk profiles provide opportunities to guide medical decision-making, not all of the predictions are actionable. To illustrate the potential clinical utility, we focused on the primary prevention of cardiovascular disease and the response to newly emerging health threats the example of COVID-19.

Risk scores are well established in the primary prevention of cardiovascular events and have been recommended to guide preventive lipid-lowering interventions[47]. While cardiovascular predictors are accessible at a low cost, dedicated visits and resources from healthcare providers for physical and laboratory measurements are required. Therefore, we compared our phenome-wide risk score, based only on age, sex, and routine health records, to models based on established cardiovascular risk scores, the SCORE2[48], the ASCVD[3], and the British QRISK3[4] score. Interestingly, the discriminative performance of our phenome-wide model is competitive with the established cardiovascular risk scores for all investigated cardiovascular endpoints (Fig. 5a and Supplementary Data 7): we found comparable C-Indices with differences −0.001 (−0.004, 0.002) for ischemic stroke, −0.003 (−0.005, −0.001) for ischemic heart disease and +0.002 (−0.001, 0.004) for myocardial infarction compared with the comprehensive QRISK3 score. It is noteworthy that these discriminative improvements are substantially better for later-stage diseases, including heart failure (+0.014 (0.012, 0.017)), cardiac arrest (+0.003 (−0.003, 0.01)), and all-cause mortality (+0.017 (0.015, 0.019)) when prior health records are considered.

To further illustrate potential utility, we look at newly emerging pathogenic health threats, where rapid and reliable risk stratification is required to protect high-risk groups and prioritize preventive interventions. We investigated how our phenome-wide risk states could have been used in the context of COVID-19, a respiratory infection with pneumonia and sepsis as common, life-threatening complications of severe cases. We repurposed the risk states for pneumonia, sepsis, and all-cause mortality to calculate a combined COVID-19 severity risk score using information available at the end of 2019 before the global spread of the COVID-19 pandemic (see "Methods" for details). The COVID-19 severity risk score resembles the risk for developing severe or fatal COVID-19 and illustrates how health records could have helped to identify individuals at high risk and to prioritize individuals in initial vaccination campaigns better. Augmenting age with the COVID-19 severity risk score, we found substantially improved discriminative performance for both severe and fatal COVID-19 outcomes (Severe: C-Index (age) 0.597 (CI 0.591, 0.604) → C-Index (age + COVID-19 severity risk score) 0.663 (CI 0.656, 0.669); Fatal: C-Index (age) 0.719 (CI 0.709, 0.728) → C-Index (age + COVID-19 severity risk score) 0.738 (CI 0.728, 0.748). These discriminative improvements translate into higher cumulative incidence in the Top 5% population compared to age alone (Fig. 5c, age (left), COVID-19 severity score (right), severe COVID-19 (top), fatal COVID-19 (bottom)): In the top 5% of the age-based risk group (~ 79 (IQR 77, 81) years old), 0.41% (CI 0.33%, 0.48%, n = 102) have been hospitalized, and 0.28% (CI 0.21%, 0.34%, n = 70) had died by the end of the first wave. By the end of the second wave, around 0.96% (CI 0.83%, 1.08%, n = 240) had been hospitalized, and 0.48% (CI 0.4%, 0.57%, n = 121) had died. In contrast, for individuals in the top 5% of the COVID-19 severity risk score, by the end of the first wave, around 0.58% (CI 0.49%, 0.68%, n = 146) had been hospitalized, and 0.4% (CI 0.32%, 0.48%, n = 100) had died, while by the end of the second wave, 1.54% (CI 1.38%, 1.69%, n = 386) had been hospitalized and 0.74% (CI 0.63%, 0.84%, n = 185) had died.

In summary, our findings illustrate the clinical utility of medical history for primary prevention of cardiovascular diseases and the rapid response to emerging health threats.

## Discussion

Current clinical practice lacks systematic, data-driven guidance for individuals and care providers. Our study demonstrated that medical history can systematically inform on phenome-wide risk across clinical specialties, as shown in the British UK Biobank cohort. Subsequently, we show that these risk states can be repurposed to identify individuals vulnerable to severe COVID-19 and mortality. Importantly, we found significant improvements in the discriminated performance for the vast majority of disease endpoints, of which almost 80% could be replicated in the US All of US cohort. Our results indicated utility beyond age, sex, selected comorbidities, and established cardiovascular risk factors commonly considered in clinical practice for preventable diseases, treatable diseases, and diseases without existing risk stratification tools. We anticipate that our approach has the potential to facilitate population health at scale.

Designed for outpatient settings and focused on patients without acute complaints, our approach identifies incident disease onset from early (e.g., hypertension) and later (e.g., bypass surgery) health system contacts. We identified three primary scenarios of potential utility: Firstly, medical history can be exploited in diseases that are preventable with effective interventions, such as the prescription of lipid-lowering medication for primary prevention of coronary heart disease[47]. Lowering LDL cholesterol in 10,000 individuals at increased risk by 2 mmol/L with atorvastatin 40 mg daily (~ 2€ per month) for 5 years is estimated to prevent 500 vascular events, reducing the individual relative risk by more than a third[49,50]. Secondly, in conditions that are not preventable anymore, individuals can benefit from early detection and treatment, like in type 2 diabetes or systolic heart failure. In individuals with heart failure with reduced ejection fraction, a comprehensive treatment regime (including ARNI, beta-blockers, MRA, and SGLT2 inhibitors) compared to a conventional regime (ACEi or ARB and beta blockers) reduced the hospital admissions for heart failure by more than two thirds, all-cause mortality by almost half[51]. For a 55-year-old male, this translated into an estimated 8.3 additional years free from cardiovascular death or readmission for heart failure. Lastly, in cases where outcomes are neither preventable nor treatable, estimates of prospective individual risk may be of high importance for personal decisions or the planning of advanced care, e.g., a high short-term mortality could identify patients needing to transition from curative to palliative strategies for optimal care[52,53]. Multiple studies have shown that palliative care services can improve patients' symptoms and life quality and may even increase survival[54]. Overall, our approach could facilitate the identification of high-risk populations for specific screening programs, potentially improving the value of national health programs.

Importantly, our approach, based on routine health records, shows large discriminative improvements for the majority of diseases compared with conventionally tested biomarkers[55–57] and can be generalized across diverse health systems, populations, and ethnicities. However, we also see that including the medical history over age and sex deteriorates the performance for a subset of 0.7% (UK Biobank) and 5.5% (All Of Us cohort), respectively. Three central challenges remain: First, health records, being products of interactions with the medical system, are subject to biological, procedural, and socio-economic biases[58], as well as being dependent on the evolving nature of medical knowledge and policies. Furthermore, certain measurements and laboratory values are often inaccessible at the point of care, and harmonization in and across health systems presents a significant barrier to implementation[59]. Integrating these measures into the model holds considerable promise to improve the predictive performance further. Importantly, while all UK Biobank participants have given explicit consent for researchers to access all their medical and health-related records, primary care records are currently only available for 229,944 individuals, and increasing this coverage would likely further improve the performance of the model. While our approach is

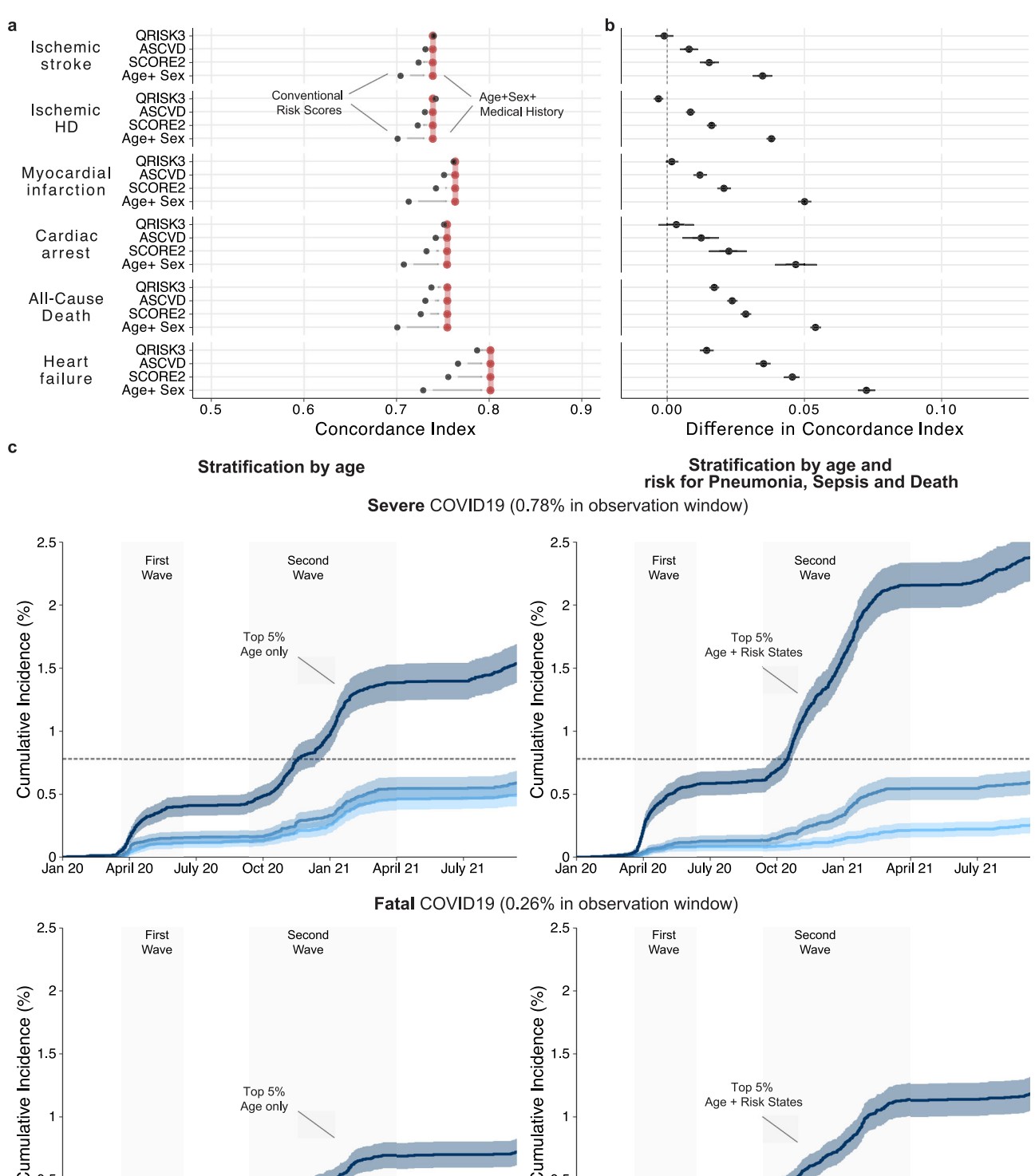

**Fig. 5 | Predictions can support cardiovascular disease prevention and the response to emerging health threats. a** Discriminatory performances in terms of absolute C-Indices comparing risk scores (Age + Sex, SCORE2, ASCVD, and QRISK as indicated) with the risk model based on Age + Sex + RiskState (red segment). **b** Direct differences between risk scores (Age + Sex, SCORE2, ASCVD, and QRISK as indicated) and the risk model based on Age + Sex + RiskState in terms of C-index. Dots indicate medians and whiskers extend to the Bonferroni-corrected 95% confidence interval for a distribution bootstrapped over 100 iterations. **c** Estimated cumulative event trajectories, including 95% confidence intervals of severe (with hospitalization) and fatal (death registry) COVID-19 outcomes stratified by the Top, Median, and Bottom 5% based on age (left), or risk states of pneumonia, sepsis and all-cause mortality as estimated by Kaplan-Meier analysis. Statistical measures were derived from 502,489 individuals. Source data are provided.

based on the standardized OMOP vocabulary, implementation requires a robust harmonization infrastructure and data drift might necessitate model updates. In addition, our findings are based on harmonization of the point-of-care vocabularies into PheCodes, and using SNOMED codes directly or other aggregation methods like ICD might yield numerically different results. Second, research cohorts often comprise healthier individuals with lower disease prevalence than the general population[60], potentially leading to underestimating absolute risks. While discriminative improvements provide evidence of the potential clinical utility, they are insufficient to prove it, as it is highly context-dependent on the population, the disease, and the interventions available. This is particularly relevant for very rare diseases, where screening the general population poses the risk of false positive findings. Future randomized implementation studies must investigate how this discriminatory information can translate into improved clinical outcomes in the respective target populations. The third challenge concerns ensuring the interpretability of our approach to such complex data. Our approach provided unique insights into how the model used patients' medical history to make risk predictions. The Shapley value attributions highlighted features the model found most informative for inference on both individual and population levels. These attributions are reflective of the model's decision-making process, and while they are aligned with our clinical understanding, they should not replace clinician judgment or other forms of evidence. As we refine and deploy this approach, we must remain vigilant in evaluating its performance and understanding the interpretational limitations. Interestingly, the attributions also expose the challenges of implementing predictive models across primary care and clinical specialties. For example, statins and chest pain are among the most highly attributed records for a high future likelihood of developing heart disease, indicating that in some cases, prior healthcare providers have already considered or even acted upon a high suspected risk of the disease, without entering the actual diagnosis into the records. Consequently, employing the model for such patients, when low-density lipoprotein (LDL) cholesterol levels are already managed, may not lead to further preventive actions if the patient's care aligns with established standards. Importantly, we find that such cases do not drive the model's predictive performance by assessing the robustness of the model performance to the removal of recent information (Supplementary Fig. 6). Ultimately, if routine health records are to be used for risk prediction, robust governance rules to protect individuals, such as opt-out and usage reports, need to be implemented. With many national initiatives emerging to curate routine health records for millions of individuals in the general population, future studies will allow us to better understand how to overcome these challenges.

Our study presents a systematic approach to simultaneous risk stratification for thousands of diseases across clinical specialties based on readily available medical history. These risk states can then be used to rapidly respond to emerging health threats such as COVID-19. Our findings demonstrate the potential to link clinical practice with already collected data to inform and guide preventive interventions, early diagnosis, and treatment of disease.

## Methods

### Data source and definitions of predictors and endpoints
To derive risk states, we analyzed data from the UK Biobank cohort. Participants were enrolled from 2006 to 2010 in 22 recruitment centers across England, Scotland, and Wales; the follow-up is ongoing, and records until the 19th of December 2022 are included in this analysis. The UK Biobank cohort comprises 273,375 women and 229,114 men aged between 37–73 years at the time of their assessment visit. Participants are linked to routinely collected records from primary care (GP, currently only available for half the cohort), hospital records (HES, PEDW, and SMR), and death registries (ONS), providing longitudinal

information on diagnosis, procedures, and prescriptions for the entire cohort from Scotland, Wales, and England. Routine health records were mapped to the OMOP CDM and represented as a 65,115-dimensional binary vector, indicating whether a concept has been recorded at least once in an individual prior to recruitment. A subset of 14,445 unique concepts, all found in at least 50 individuals, was chosen for model development. Endpoints were defined as the set of PheCodes X[39,61], and after the exclusion of very rare endpoints (recorded in < 100 individuals), 1741 PheCodes X endpoints were included in the development of the models. Due to the adult population, congenital, developmental, and neonatal endpoints were excluded. For each endpoint, subsequently, time-to-event outcomes were extracted, defined by the first occurrence after recruitment in primary care, hospital, or death records. Detailed information on the predictors and endpoints is provided in Supplementary Data 1, 2.

While all individuals in the UK Biobank were used to integrate the routine health records, develop the model, and estimate phenome-wide log partial hazards, individuals were excluded from endpoint-specific downstream analysis if they were already diagnosed with a disease (defined by a prior record of the respective endpoint) or are generally not eligible for the specific endpoint (females were excluded from the risk estimation for prostate cancer).

To externally validate our risk states, we investigate individuals from the All of Us cohort[37], containing information on 259,234 individuals of diverse descent and from minorities historically underrepresented in biomedical research[40]. Because we only use the All of Us cohort for validation, we evaluate the predictive performance for the subset of 1519 endpoints with at least 100 incident events in the All of Us cohort.

The study adhered to the TRIPOD (Transparent Reporting of a Multivariable Prediction Model for Individual Prognosis Or Diagnosis) statement for reporting[62]. The completed checklist can be found in the Supplementary Information.

### Extraction and preparation of the routine health records
To extract the routine health records of each individual, we first aggregated the linked primary care, hospital records, and mortality records and mapped the aggregated records to the OMOP CDM (mostly SNOMED and RxNorm). Specifically, we used mapping tables provided by the UK Biobank, the OHDSI community, and SNOMED International to map concepts from the provider and country-specific non-standard vocabularies to OMOP standard vocabularies.

We restricted the analysis to the domains "Measurement", "Observation", "Condition", "Procedure", "Drug" and "Device". To reduce the complexity, we included laboratory measures only as binary indicators, i.e., whether a measurement was taken, but not the measurement value. The PheCode X endpoints[39,61] were derived from either mapping directly from ICD-10 (hospital and death records) or mapping from SNOMED to ICD-10 (using the official mapping table) and subsequently to Phecodes X.

To ensure the accuracy and integrity of our data, we implemented multiple validation steps. After each stage in the extraction and mapping process, we conducted plausibility and sanity checks on the distribution of the mapped records, along with spot checks of individual records. This approach was critical in verifying the validity of the data. In addition, post-model training, the data underwent further verification. This included analyzing the calculated record attributions and removing recent records, as detailed in Supplementary Fig. 6. These steps were essential to identify and mitigate any potential issues of record leakage. In the accompanying code release, we have provided the exact code used to extract and prepare the health records.

### Spatial validation and data preprocessing
For model development and testing, we split the data set into 22 spatially separated partitions based on the location of the assessment

center at recruitment. We analyzed the data in 22-fold nested cross-validation, setting aside one of the spatially separated partitions as a test set, aggregating the remaining partitions, and randomly selecting 10% of the aggregated data for the validation set. Within each of the 22 cross-validation loops, the individual test set (i.e., the spatially separated partition) remained untouched throughout model development, and the validation set was used to validate the fitting progress and checkpoint selection. All 22 obtained models were then evaluated on their respective test sets. We assumed missing data occurred randomly and performed multiple imputations using chained equations with gradient boosting machines[63,64]. Imputation models were fitted on the training sets and applied to the respective validation and test sets. Continuous variables were standardized; Categorical variables were one-hot encoded.

## Development of the phenome-wide risk model

The risk model is a multi-task neural network that uses the binary representations of an individual's prior health records before recruitment to simultaneously predict log partial hazards[65] for a set of 1741 endpoints. The model consists of three fully connected linear layers with 4096 hidden units, each with layer normalization[66], dropout[67], and leaky ReLU activations. The last latent representation serves as a regulariser as it incentives the extraction of robust features for multiple diseases. For comparison, we also benchmarked the linear version of our model with 25.2 M instead of 82.3 M parameters (see Supplementary Fig. 2). The model subsequently computes the log partial hazard (the risk state) for each endpoint with an adapted proportional hazard loss[65], resulting in a 1741-dimensional output representation. The individual losses are averaged and then summed to derive the final loss of the model. We subsequently tuned hyperparameters (via Bayesian Optimization) on train and validation splits over a constrained parameter space, tuning batch size, learning rate, weight decay, number of nodes in the layers of the endpoint heads, number of hidden layers, dropout rates, and size of the output vector of the shared network. The final models were trained with batch size 512 using the Adam optimizer[68] with a learning rate of 0.0006 and weight decay of 0.3, and early stopping tracking of the performance on the validation set. We implemented the model in Python 3.9 using PyTorch 1.11[69] and PyTorch-lightning 1.5.5 (for code availability, see below). The training of a single model on an NVIDIA A100 GPU node for 18 epochs required approximately 11 h, equating to the emission of approximately 1.08 kg $CO_2$ eq, 4.36 km driven by an average ICE car or 0.54 kgs of coal burned as calculated by the mlco2 calculator[70]. The external validation of these models, conducted within the All of Us cloud computing environment and including data preprocessing, inference, and evaluation, incurred a total compute cost of approximately 150 USD.

## Downstream analysis and performance comparisons

We fitted Cox proportional hazards models[36] (CPH) to derive absolute risk predictions from the endpoint-specific risk states for the individual endpoints. For each endpoint, we developed models with distinct covariate sets: For all endpoints, we investigated age, biological sex, and the risk states from the health records. For cardiovascular endpoints, we additionally investigated predictors from established and guideline-recommended scores for the primary prevention of cardiovascular diseases, the SCORE2, ASCVD, and QRISK3. Model development was repeated independently for each assessment center thus, for each cross-validation split, models were trained on the respective train set, and checkpoints were selected on the respective validation set. For the final evaluation, test set predictions from the spatially separate recruitment centers were aggregated. Event risk rates were calculated over the full observation period. Harell's C-Index[71] was calculated with the lifelines package[72] by bootstrapping both the aggregated test set and individual assessment centers within ten years after recruitment to control for right-censoring. The C-Index is a measure of rank correlation that quantifies the agreement between predicted and observed outcomes. It ranges between 0.5 (no better than random prediction) to 1 (perfect prediction). Statistical inferences about model differences were based on the distribution of bootstrapped differences in the C-Index; models were considered different whenever the Bonferroni-corrected 95% CI of the difference did not overlap cross zero, to account for multiple testing. CPH models were fitted with the CoxPHFitter from the Python package lifelines[72] with default parameters and a step size of 0.5, 0.1, or 0.01 to facilitate model convergence. Confidence intervals for all statistical analyses were calculated over 1000 bootstrapping iterations.

## Response to emerging health threats

We retrained our models using data limited to records until the end of December 2019, keeping the setting (in particular time zero for training) unchanged. Using these updated models, we then predicted the risk states using all data available at the end of 2019, just as the first cases of COVID-19 were reported. We then manually selected specific risk states associated with pneumonia, sepsis, and all-cause mortality to create an unweighted COVID-19 severity risk score. This risk score was subsequently tested against age for the identification of incident severe and fatal COVID-19 cases.

## Independent validation in the All Of Us cohort

After mapping the linked health records from All Of Us to the OMOP vocabulary, we transferred the neural networks developed in the UK Biobank to the All Of Us research environment. We then used the models to predict the disease-specific risk states for all individuals. Subsequently, we predicted absolute risks with the CPH models developed in the UK Biobank. Finally, we calculated the mean of the predictions from the models for each individual and disease. For baseline comparison with Age and Sex, we fitted new CPH models in the All Of Us cohort.

As part of validating our risk model, we tested its performance across diverse groups included in the All Of Us cohort, which focuses on populations historically underrepresented in biomedical research. These groups include individuals identifying with a single race other than White, multiple races, or non-White ethnic backgrounds, such as Japanese. The cohort also includes older adults aged 65 or older and sexual and gender minorities, such as people who are intersex, non-binary, transgender, or whose gender identity differs from their sex at birth, as well as those with sexual orientations other than straight, such as gay, lesbian, or bisexual. It encompasses individuals with lower household incomes (at or below 200% of the Federal Poverty Level), those with less formal education (without a high school diploma or GED), residents of rural or non-metropolitan areas, people with disabilities that limit daily activities, and those with limited access to healthcare, such as individuals without insurance or primary care.

## Calculation of record attributions

To determine which records are most important on an individual level, we calculated attributions for the selection of 24 endpoints based on Shapley values. For computational efficiency, we approximated Shapley values via sampling for 9460 individuals unseen to the model during development[43]. Please refer to Supplementary Data 9 for the aggregated attributions from individuals without prior events. Shapley values in the table are provided in two forms: averaged (so called local attributions to quantify importance for affected individuals) and summed (global attributions to quantify importance for population ranking). The average Shapley attributions, presented in the main text and figures, closely reflect our understanding of the importance of affected individuals.

## Reporting summary

Further information on research design is available in the Nature Portfolio Reporting Summary linked to this article.

## Data availability

UK Biobank data, including all linked routine health records, are publicly available to bona fide researchers upon application at http://www.ukbiobank.ac.uk/using-the-resource/. In this study, only primary care data not subject to the Government's Control of Patient Information (COPI) notice was used (UK Biobank Category 3000). The All Of Us cohort data were provided by the All Of Us Research Program by permission that can be sought by scientists and researchers whose institutions have signed a Data Use and Registration Agreement (DURA). All patient data used throughout this study has been subject to patient consent as covered by the UK Biobank and All Of Us. Detailed information on the predictors and endpoints is presented in Supplementary Data 1–3. Source data are provided in this paper.

## Code availability

All code developed and used throughout this study has been made open source and is available on GitHub. The code to train the medical history model can be found here: github.com/nebw/medhist, while the code to run analysis on trained models can be found here: github.com/JakobSteinfeldt/MedicalHistoryPhenomeWide.

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

## Acknowledgements

We would like to acknowledge the support of the UK Biobank and the All of Us Research Program in providing access to their respective datasets. This research has been conducted using data from the UK Biobank (application number 49966 and 51157) and the All of Us Research Program (by S.H. UserID 5703, and B.W. UserID benwild@researchallofus.org). Both studies have received ethical approval from their respective institutional review boards and have obtained informed consent from participants. We are grateful to the participants who generously contributed their time and data to make this research possible. This project has been funded by the Charité - Universitätsmedizin Berlin and the Einstein Foundation Berlin through the Einstein BIH Visiting Fellowship awarded to J.D. The study has been supported by the BMBF-funded Medical Informatics Initiative (HiGHmed, 01ZZ1802A - 01ZZ1802Z) and the Deutsche Forschungsgemeinschaft (DFG, German Research Foundation) – Project-ID 437531118 – SFB 1470. S.D. is supported by (a) the BHF Data Science Center led by HDR UK (grant SP/19/3/34678), (b) BigData@Heart Consortium, funded by the Innovative Medicines Initiative-2 Joint Undertaking under grant agreement 116074, (c) the NIHR Biomedical Research Center at University College London Hospital NHS Trust (UCLH BRC), (d) a BHF Accelerator Award (AA/18/6/24223), (e) the CVD-COVID-UK/COVID-IMPACT consortium and (f) the Multimorbidity Mechanism and Therapeutic Research Collaborative (MMTRC, grant number MR/V033867/1). HH is supported by Health Data Research UK and the National Institute for Health Research, Biomedical Research Center at University College London Hospitals.

## Author contributions

J.S., B.W., T.B., M.P., H.H., C.L., U.L., J.D., and R.E. conceived and designed the project. J.S., B.W., and T.B. implemented models, conducted experiments, and performed data analysis. J.U. and A.V. supported the analysis. S.H. performed the external validation. M.P., S.D., H.H., and C.L. provided methodological support and contributed to the discussion of the results. J.S., B.W., T.B., U.L., J.D., and R.E. wrote and prepared the manuscript. All authors read, revised, and approved the manuscript.

## Funding

## Competing interests

U.L. received research grants to the institution from Abbott, Amgen, Bayer and Novartis. J.D. received honoraria from Amgen, Boehringer Ingelheim, Merck, Pfizer, Aegerion, Novartis, Sanofi, Takeda, Novo Nordisk, Bayer, and is a Trustee of Our Future Health. R.E. received honoraria from Sanofi and consulting fees from Boehringer Ingelheim. All other authors do declare no competing interests.
