## [Transparent Peer Review file · Nature Communications]

Medical history predicts phenome-wide disease onset and enables the rapid response to emerging health threats

Corresponding Author: Roland Eils

Version 0:

Reviewer comments:

Reviewer #1

(Remarks to the Author)

This represents a technical revision to a previously published study where a (UK Biobank Covid Primary care) dataset had to be removed and replaced for regulatory compliance reasons.

The overall structure, contents of the study largely remain the same, with slight variations to reflect the updated dataset. The overall meaning of the study remains intact that broad (electronic) medical record data can be used to predict a wide range of disease states and trajectories.

Assuming all of the figures have also been updated to reflect the updated dataset, I would not have additional major revisions to recommend.

(Remarks on code availability)

Code appears to be available on a public GitHub repository with an overview Readme statement, but I have not reviewed in detail how usable or reproducible the analysis is.

Reviewer #2

(Remarks to the Author)

Overall:

This paper seems novel to this reviewer and their findings are intriguing with the use of All of Us being rather timely as that is a recently released dataset (the UK Biobank part seems a bit old and less novel). Further details that complete the Nature Review criteria and also my own comments organized into major and minor sections are as follows.

- A. Summary of the key results: Interesting findings and nice that it is validated in two different populations.
- B. Originality and significance: if not novel, please include reference: The work appears novel to this reviewer.
- C. Data & methodology: validity of approach, quality of data, quality of presentation. No complaints appears solid.
- D. Appropriate use of statistics and treatment of uncertainties: However, sexual and gender minorities is not described in the paper so its unclear what is included or what that means (e.g., is that female sex? and what if females are the majority in one dataset and the minority in the other dataset?).
- E. Conclusions: robustness, validity, reliability. This paper's findings are neat and interesting.
- F. Suggested improvements: experiments, data for possible revision: Those are provided subsequently and organized as major and minor.
- G. References: appropriate credit to previous work?: Seemed appropriate.
- H. Clarity and context: lucidity of abstract/summary, appropriateness of abstract, introduction and conclusions: Seemed appropriate..

Minor comments:

1. The phrase 'sexual and gender minorities' is used in figure 3 but should also be defined in the figure legend mention in the text or figure legends. What does this mean in the context of this study, is this females? Is this males? Is this females and non-binary? This should be defined because the 'minority' sex or gender may vary from dataset to dataset (depends on data

collection methods and other issues).

2. Limitations of PheCodes as 'phenotypes' should be discussed. Other methods of grouping meaningful codes together might yield different results (e.g., the use of SNOMED), this should be discussed in the discussion.

(Remarks on code availability)

Page was not found - Code was unavailable

Response to the Reviewers

REVIEWERS' COMMENTS

Reviewer #1 (Remarks to the Author):

Remarks to the Author	Answer
This represents a technical revision to a previously published study where a (UK Biobank Covid Primary care) dataset had to be removed and replaced for regulatory compliance reasons. The overall structure, contents of the study largely remain the same, with slight variations to reflect the updated dataset. The overall meaning of the study remains intact that broad (electronic) medical record data can be used to predict a wide range of disease states and trajectories. Assuming all of the figures have also been updated to reflect the updated dataset, I would not have additional major revisions to recommend.	We thank the reviewer for their positive feedback.
Code appears to be available on a public GitHub repository with an overview Readme statement, but I have not reviewed in detail how usable or reproducible the analysis is.	We thank the reviewer for this positive feedback. Please find the corresponding code referenced under 'Code availability' in the manuscript and available in github under this link: https://github.com/JakobSteinfeldt/MedicalHistoryPhenomeWide

Reviewer #2 (Remarks to the Author):

Remarks to the Author	Answer
Overall: This paper seems novel to this reviewer and their findings are intriguing with the use of All of Us being rather timely as that is a	We thank the reviewer for the positive feedback on our resubmitted paper.

recently released dataset (the UK Biobank part seems a bit old and less novel). Further details that complete the Nature Review criteria and also my own comments organized into major and minor sections are as follows.

- A. Summary of the key results: Interesting findings and nice that it is validated in two different populations.
- B. Originality and significance: if not novel, please include reference: The work appears novel to this reviewer.
- C. Data & methodology: validity of approach, quality of data, quality of presentation. No complaints appears solid.
- D. Appropriate use of statistics and treatment of uncertainties: However, sexual and gender minorities is not described in the paper so its unclear what is included or what that means (e.g., is that female sex? and what if females are the majority in one dataset and the minority in the other dataset?).
- E. Conclusions: robustness, validity, reliability. This paper's findings are neat and interesting.
- F. Suggested improvements: experiments, data for possible revision: Those are provided subsequently and organized as major and minor.
- G. References: appropriate credit to previous work?: Seemed appropriate.
- H. Clarity and context: lucidity of abstract/summary, appropriateness of abstract, introduction and conclusions: Seemed appropriate.

Minor comments:

1. The phrase 'sexual and gender minorities' is used in figure 3 but should also be defined in the figure legend mention in the text or figure legends. What does this mean in the context of this study, is this females? Is this males? Is this females and non-binary? This should be defined

We thank the reviewer for this important comment and request for clarification. Sex and gender minorities in particular refers to people who select intersex as their sex at birth, select non-straight as their sexual orientation or select any gender identity choice other than man or women. Based on the comment, we now added the definitions from the corresponding citation to the methods.

because the 'minority' sex or gender may vary from dataset to dataset (depends on data collection methods and other issues).	
2. Limitations of PheCodes as 'phenotypes' should be discussed. Other methods of grouping meaningful codes together might yield different results (e.g., the use of SNOMED), this should be discussed in the discussion.	We thank the reviewer for this important comment and added the limitation as recommended to the discussion.
Page was not found - Code was unavailable	We thank the reviewer for this feedback. Please find the corresponding code referenced under 'Code availability' in the manuscript and available in github under this link: https://github.com/JakobSteinfeldt/MedicalHistoryPhenomeWide